# Survival Analysis of Oncological Patients Using Machine Learning Method

**DOI:** 10.3390/healthcare11010080

**Published:** 2022-12-27

**Authors:** Latefa Hamad Al Fryan, Malik Bader Alazzam

**Affiliations:** 1Department of Educational Technology, College of Education, Princess Nourah bint Abdulrahman University, Riyadh 11671, Saudi Arabia; 2Department of Medical Instruments Engineering Techniques, Al-Farahidi University, Baghdad 10021, Iraq

**Keywords:** data mining machine learning, cancer, hospital cancer registries, Baghdad Teaching Hospital, survival analysis

## Abstract

Currently, a considerable volume of information is collected and stored by large health institutions. These data come from medical records and hospital records, and the Hospital Cancer Registry is a database for integrating data from hospitals throughout Iraq. The data mining (DM) technique provides knowledge previously not visible in the database and can be used to predict trends or describe characteristics of the past. DM methods can include classification, generalisation, characterisation, clustering, association, evolution, pattern discovery, data visualisation, and rule-guided mining techniques to perform survival analyses that take into account all the patient’s medical record variables. For four of the eleven groups examined, this accuracy was relatively high. The database of patients treated by the Baghdad Teaching Hospital between 2018 and 2021 was examined using a classification of the most crucial variables for event prediction, and a distinctive pattern was found. Machine learning techniques allow a global assessment of the data that is available and produce results that can be interpreted as significant information for epidemiological studies, even in cases where the sample is small and there is a lack of information on several variables.

## 1. Introduction

Oncology is a branch of medicine that focuses on the early identification, treatment, and prevention of cancer. Cancer arises when abnormal cells proliferate and spread unchecked throughout the body. Recent advances in genetics and biology have revolutionized cancer therapy. Nowadays, many cancer types are treatable. An oncologist is in charge of a multidisciplinary care team that typically consists of surgeons, radiologists, pathologists, nurses, and social workers [1]. These teams are dedicated to providing patients with cancer with the care that is supported by research. Oncology, a division of internal medicine, is often used interchangeably with medical oncology. Other types of oncology include gynecologic oncology, which focuses on the medical and surgical treatment of cancer affecting the female reproductive organs, pediatric hematology and oncology, which provides medical care for babies and children with cancer, radiation oncology, which focuses on radiation treatment for cancer, and surgical oncology, which utilizes surgery to diagnose and treat a range of malignancies [2]. For the two years between 2018 and 2021, thousands of new cases of cancer are anticipated in Iraq [3]. The Hospital Cancer Registries are a source of systematically arranged and publicly accessible clinical data, and they represent a potential target for data mining and pattern discovery for the disease in Iraq.

Survival analyses involve observing events as they change over time, with the most common event being death. The Kaplan-Meier statistical method, which was originally developed to measure the frequency or number of patients who survived a given medical treatment, is used to carry them out. [4] The probability of survival over time is calculated in this manner by comparing the effects of various factors on patient survival. Large health institutions currently collect and store a significant amount of data. These data are derived from medical records and hospital records, and the Hospital Cancer Registry is a database that collects information from hospitals across Iraq [5]. The amount of information this system provides necessitates a more robust and adequate analysis to generate useful and quality knowledge from these data. The data mining technique reveals previously hidden knowledge in the database and can be used to forecast trends or describe historical characteristics. Classification, generalisation, characterisation, clustering, association, evolution, pattern discovery, data visualisation, and rule-guided mining techniques are examples of DM methods [6].

The objective of applying DM is the process of discovering knowledge in databases or Knowledge Discovery in Databases (KDD); the definition of applying DM is given by Fayyad et al. (1996) [7] as “the non-trivial process of identifying valid patterns, unknown, potentially useful, and ultimately understandable data”. Currently, the application of data mining techniques makes it possible to extract useful information from complex biological data and is directly related to the area of bioinformatics, whose main objective is to integrate and study any biological data in a strategy through information technology and computer science [8].

Regarding the development of new technologies from this growing area, it is possible to observe, in recent searches, the prioritisation of process optimisation, such as diagnostic imaging, classification of patients into groups of risk, risk prediction in transfusion, and identification of new risk factors for diseases such as cancer and diabetes [9]. In addition to these applications, the algorithms are also widely applied to discover potential new drug combinations for the treatment of diseases such as cancer and Parkinson’s disease [10].

In the current study, survival analyses based on clinical data from the Hospital Cancer Registry of patients treated at Baghdad Teaching Hospital (BTH), Iraq, were conducted using a data mining program to analyze data from the Hospital Cancer Registry. The Hospital Cancer Registry is a database for integrating data, and these data are derived from medical records and hospital records. Machine learning algorithms enable an overall evaluation of available data and provide outcomes that may be viewed as important data for epidemiological investigations.

The paper is organized as follows: Section 1.1 is Theoretical Analysis, Section 1.2 is Related Works, Section 2 shows Materials and Methods, and Section 3 shows the Results and Discussion. Finally, Section 4 is the work's Conclusion.

### 1.1. Theoretical Analysis

The main goals of the Random Forest (RF) algorithm, which was developed by Leo Breiman in 2001 [11], are to solve two different types of problems: to access and categorise variables based on their propensity to predict a particular response and to build a prediction rule in a supervised learning problem. The final prediction is the result of combining the predictions made in each decision tree that was created during the process, each of which was slightly different. A string of straightforward questions with “yes” or “no” responses yield each node of the decision tree. The risk assessment for procedures like blood transfusions is done using a free web interface, and tools that attempt to predict risk use prediction models based on Random Forest [12]. Its function is also present in immunohistochemical analyses, where it was possible to categorise parameters discovered through the study of histone modification patterns to foretell the recurrence of tumors.

In recent studies by Hamidi et al. (2016) [13], the Random Survival Forest (RSF) method was used, based on the RF, to identify risk factors in the kidney transplant process, showing a better result for the RSF than for conventional models. The use of the RF method compared to the RSF is cited in the literature as being used to predict overall survival, distant metastasis, and local recurrence of rectal cancer.

In addition to serving as a basis for implementing other models, using the RF method, a unified model for analysis of survival, regression, and classification (RF-SRC: Random forests for Survival, Regression, and Classification) was also implemented, available in R and created by authors Udaya Kogalur and Hemant Ishwaran (2016) [14]. For survival analyses, the terminal nodes of the created trees give answers based on the Kaplan-Meier (KM) estimate, which in turn predicts a probability of survival over a given period [15].

Survival data are normally analysed using methods that use restricted hypotheses that assume risks proportional to a given assumption. The problem with such methods is the requirement for parametric data; in addition, these methods present difficulty in analysing the interaction of multiple variables. The RF is a completely non-parametric statistical method which does not require a normal distribution for data analysis but still offers efficiency in the analysis of different data sets. This model can optimise the analysis of each variable’s influence on survival, both in isolation and in association with other variables.

### 1.2. Related Works

“Deep learning-based breast cancer grading and survival analysis on whole-slide histopathology images”.

A deep learning-based breast cancer grading model that utilizes whole-slide histopathology pictures was developed and tested in this study. The model was created utilizing whole-slide photos of 706 young (<40 years) patients with invasive breast cancer and the accompanying tumor grade (low/intermediate vs. high), as well as the tumor grade’s component nuclear grade, tubule formation, and mitotic rate. Using expert pathologists’ annotations as the ground truth, Cohen’s kappa was used to assess the model’s performance on a separate test set of 686 cases. Survival analysis was conducted using the projected low/intermediate (*n* = 327) and high (*n* = 359) grade groups. In comparison to skilled pathologists, the deep learning system differentiated between low/intermediate versus high tumor grade with a Cohen’s kappa of 0.59 (80% accuracy). The two groups predicted by the method were shown to have substantially different overall survival (OS) and disease/recurrence-free survival (DRFS/RFS) (*p*  <  0.05) in the subsequent survival study. Statistically significant hazard ratios (*p*  <  0.05) were shown using a single-variate Cox hazard regression analysis. The hazard ratios exhibited a trend but lost statistical significance for each outcome once stratification for molecular subtype and clinicopathologic characteristics were taken into account. Finally, a deep learning-based model for automatically assessing breast cancer on whole-slide photos was constructed. The model shows a trend in the survival of the two projected groups and distinguishes between low/intermediate and high-grade malignancies [16].

2.“Fast training of support vector machines for survival analysis”.

The study offered three different linear survival support vector machine training algorithms: ranking-based, regression-based, and combined ranking and regression. To reduce training’s processing expenses, optimization in the primal was carried out utilizing reduced Newton optimization and order statistic trees. We use the same optimization method and broaden it to include non-linear models. Our findings show that our suggested optimization technique outperforms conventional training algorithms, which struggle when applied to huge datasets because of their inherent high time and spatial complexity. On six real-world datasets, the suggested survival models were validated, and the results demonstrate that pure ranking-based techniques perform better than regression and hybrid models [17].

3.“Prediction of survival and metastasis in breast cancer patients using machine learning classifiers”.

To predict BC survival and metastasis, this study compared the efficacy of six machine-learning algorithms with two established methodologies. We utilized a dataset that contained the medical histories of 550 people with breast cancer. For the prediction of breast cancer survival and metastasis, Naive Bayes (NB), Random Forest (RF), AdaBoost, Support Vector Machine (SVM), Least-square SVM (LSSVM), Adabag, Logistic Regression (LR), and Linear Discriminant Analysis were employed.

The findings reveal 550 patients were included, and 85% of them were still living and had no metastases. The SVM and LDA have higher sensitivity (73%) in contrast to other approaches when it comes to predicting survival, with an overall average specificity of all techniques of ≥94%. The SVM and LDA have a higher overall accuracy (93%) than other algorithms. The RF had the best specificity (98%), the NB had the most sensitivity (36%), and the LR and LDA had the highest overall accuracy (86%) for predicting metastasis. Our results demonstrated that the SVM performed better than other machine learning techniques in predicting the patients’ survival in terms of several factors [18].

4.“Convolutional Neural Networks and Support Vector Machines for Five-Year Survival Analysis of Metastatic Rectal Cancer”.

The study used metastatic immunohistochemical samples stained for the protein RhoB to evaluate the usage of pre-trained convolutional neural networks and support vector machines for predicting the survival rate of a cohort of rectal cancer patients. In most situations, especially in those where the manual pathological diagnosis is very challenging, the combination of convolutional neural networks with support vector machines produced better classification results than utilizing individual pre-trained deep networks. ResNet-101 and SVM in particular provided an average accuracy of 86% for non-radiotherapy, whereas Inception-v3 and SVM produced an average accuracy of 85% for radiotherapy [19].

5.“Prognostic Nomograms for Predicting Overall Survival and Cancer-Specific Survival in Patients with Head and Neck Mucosal Melanoma”.

The purpose of the research was to develop and verify a reliable predictive nomogram for HNMM. To build the model, HNMM patients who had surgery between 2010 and 2015 were chosen from the Surveillance, Epidemiology, and End Results (SEER) database. Finally, 288 individuals were selected and randomly assigned to a training group (199 cases) and a validation cohort after removing invalid and missing clinical data (54 cases). The training cohort underwent univariate and multivariate Cox proportional hazards regression analysis to find prognostic factors. To construct the model, independent influencing variables were considered. The concordance indices (C-indices) and calibration curves were utilized to assess the prediction power of the nomogram via internal verification (training cohort) and external verification (validation cohort). The training cohort’s five independent risk factors—age, location, T stage, N stage, and surgery—were chosen based on the results and nomograms with predicted cancer-specific survival (CSS) and overall survival (OS) at 1 and 3 years were developed. Through the training queue: OS: 0.764 vs. 0.683, CSS: 0.783 vs. 0.705, the C-index demonstrated that the nomogram’s predictive performance was superior to that of the TNM staging system. This was both internally and externally validated (through the verification queue: OS: 0.808 vs. 0.644, CSS: 0.823 vs. 0.648). The calibration curves also demonstrated excellent agreement between the nomogram-based prediction and the actual survival rate. The nomogram prediction model may be useful for directing therapeutic therapy since it may predict the prognosis of HNMM patients more accurately than the conventional TNM staging approach [20].

### 1.3. Important Notes on Previous Studies

The previous studies focus on deep learning-based breast cancer grading and survival analysis by whole-slide histopathology pictures and focus on three different linear survival support vector machine training algorithms.The previous studies developed and verified a reliable predictive nomogram for HMM, built the model, and demonstrated the SVM performed better than other machine learning techniques in predicting the patient’s survival.

### 1.4. The Current Study Differs from Previous Studies

The data come from medical records and hospital records.The data mining (DM) technique was used to predict trends or describe characteristics of the past. DM methods can include classification, generalisation, characterisation, clustering, association, evolution, pattern discovery, data visualisation, and rule-guided mining techniques.The database of patients treated by the Baghdad Teaching Hospital between 2018 and 2021 was examined using a classification of the most crucial variables for event prediction.The current study uses machine learning techniques for a global assessment of the data that is available and produces results that can be interpreted as significant information for epidemiological studies.

## 2. Materials and Methods

### 2.1. Data Analysis

All data submitted for analysis were organised in spreadsheets with standardised variables and without removing fields with missing information. The types of cancer were grouped according to the International Classification of Diseases (ICD-10) [21], and the period covered by the analysis was from 2018 to 2021, using only data from patients treated at the BTH.

The following fields from the Hospital Cancer Registry were included: age, sex, occupation, treatment, tumor staging (TNM), pathological tumor staging (pTNM), histological type, alcohol and tobacco consumption history, family history of cancer, and primary location of the tumor. Patients were grouped by type of cancer, with groups with less than 50 patients being discarded, leaving only the following groups: head, face, and neck; colon and rectum; cervix; stomach; liver; lymph nodes; breast; pancreas; skin; prostate; lung. In total, the groups totaled 5877.

The clinicopathological variables mainly include tumor size and tumor grade. In addition to them we have used the variables represented: age, first treatment (prim treatment), progression of treatment (prog_treatment), pathological TNM classification (pTNM), disease staging, TNM classification of malignant tumors (TNM), patient occupation, history of smoking, sex, race or color, histology of the primary tumor (histology), history of alcoholism, tumor laterality, the basis used for diagnosis (base_diag), family history of cancer (hist_family), metastasis distance (met_dist), patient education (education), and primary tumor location. Other ML parameters include number of extracted features, valuable samples, and size of the dataset. As per the parameters for data augmentation, it includes image rotated, shifted (left or right), scaled, stretched, one axis on certain angle (shear angle), shifted vertically, brightness of the image, and flipped horizontally and vertically. Among the computation variables it includes prediction error rate, average Brier Score [22,23], and probability of survival.

Three distinct ratios of the training, test, and validation set, namely (75%, 20%, and 5%); (80%, 10%, and 10%); and (75%, 15%, and 10%), were used in our tests. It has been noted that the ratio set of (75%, 20%, and 5%) has produced the best outcomes. A total of 20 leaves were present. The Minkowski distance metric was applied to the tree. For k neighbour inquiries, ten neighbours were used. The Manhattan distance would be used if *p* = 1, which is the power parameter for the Minkowski metric. Each neighbourhood’s points were given equal weight.

### 2.2. Survival Analysis Using Machine Learning Algorithms

The randomForestSRC library available on the R Studio platform (R 3.4.4 version) [Available at: cran.r-project.org/web/packages/randomForestSRC/randomForestSRC.pdf, accessed on 5 November 2022] was used to analyse the Hospital Cancer Registry database, to classify the variables in order of relevance for survival prediction, and, in addition, to evaluate the algorithm’s performance in the prediction process. The classification of variables was quantified by the vimp (variable importance), a numerical value that corresponds to the value that the removal of the variable would add to the error rate if it is classified as of positive importance or the value that its removal would decrease the rate of error if it is classified as of negative importance.

For all groups, a pattern of creation of 1000 trees was configured using log-rank splitting and including fields with missing information through adaptive imputation of the tree. The error rate was evaluated using the Brier Score (the index that varies from 0 to 1), where 0 indicates a perfectly correct prediction and the prediction error rate according to the number of trees. The period was represented in days in the survival analyses for a total of 365 days.

### 2.3. Methodology

In the proposed algorithm, firstly, load and import the dataset and then split it into the three parts, i.e., training, test, and validation set. Then, resize all images on the same scale. After that, apply augmentation techniques on the training dataset to increase the number of images in the data set for model training. The ratio of training, test, and validation images are same before and after the augmentation step. Now, load and import the deep learning model. Then, apply the optimization function. This is referred to as the proposed model for training. After that, generate or train a model for results. Figure 1 below presents the block function used in the proposed model.

### 2.4. Ethical Aspects

The clinical data used are available at the Integrator of the BTH-Hospital Cancer Registry and do not have patient identification. The results presented also do not allow the individual identification of any patient included in the study.

## 3. Results and Discussion

### 3.1. Prediction Model Error Rate

Machine learning techniques such as Support Vector Machine (SVM) [24], 1-Nearest Neighbor (1NN), Multilayer Perception (MLP), and Random Forest (RF) [25] are widely used to develop tools for diagnosis or medical decision-making. In all cases, it is important to evaluate the accuracy of the technique, where with the same dataset different techniques may present different performances. This paper explores deep learning-based cancer grading, and RandomForest approach is used for the prediction of survival rate. The type of cancer is classified based on the grade, as shown in Table 1, as it is time consuming and tiresome to detect cancer via manual segmentation of tumors. As per the dataset, we have now included the dataset type; the dataset classifies the tumor based on its grade. The images describe the tumor type in the experiments, which is known as the grade of tumor. It is also known as multi-modeling feature of the deep learning models, which is denoted by inbuilt function.

The results are shown in Table 1, which compares the different error rates among all groups of patients present in the database; the lowest rate was 7.11% for the skin cancer group, and the highest rate was 65% for the cancer group of the liver.

Feature extraction can accurately classify tumors with the help of an accurate prediction model. This paper uses a prediction model for accuracy measurement that is based on several constraints such as ML parameters, number of extracted features, valuable samples, and size of the dataset. Several ML approaches can develop an accurate prediction model, depending on the accuracy, such as logistic regression, ensemble, tree, linear discriminant, KNN, and SVM. The accuracy improved by using the training techniques with ML approaches such as 2D deep features, histogram distribution, intensity, and statistical texture features, and location/volumetric features.

In our experiments we used an architecture by incorporating a fixed number of layers as an input layer, each of which contains the images extracted from the previous data processing stage. It has the activation functions that pass through the convolution layers. This step helps in preventing overfitting by using a dropout layer followed by a softmax layer as well as a fully connected layer. It estimates the output and finally predicts the class by classification layer. Therefore, we need to give class of input for prediction. In the case of RF, it is possible to evaluate the error rate in the prediction by comparing the predictions made by the algorithm in the training group with the data set unknown by the program (OOB group), in addition to evaluating the value of the Brier Score.

The Brier index is a measure that varies from 0 to 1 and is used to evaluate predictions of binary responses, where 0 is the value corresponding to a perfect prediction of the result; the closer to 1, the lower the accuracy of the evaluated model [22]. Among the observed values, most were close to 0, with emphasis on the skin and cervical cancer groups, where the value was on average 0.01 and 0.09, respectively. However, the two groups present discrepant sampling sizes.

The different attributes of each set of patients contribute to the difference between the error rates, since the amount of missing data in the input variables is high in some groups, such as pancreatic, liver, stomach, and lung cancer. When using variables with missing data, the calculated risk for the patient may be greater than the calculated risk based on available data, compromising the correct prediction [26].

Performance is also improved when different types of data are entered in addition to those from clinical records, such as data regarding the drugs used, genetic variables, and laboratory test images, among other relevant information [27]. To achieve more robust results, it is necessary to structure a clinical database related to the various variables associated with cancer, a disease considered multifactorial.

Figure 2 shows four different regions (confusion matrix) to provide the extraction of ML model’s feature vector. A confusion matrix is also a performance measurement technique for machine learning classification. A confusion matrix goes deeper than classification accuracy by showing the correct and incorrect (i.e., true or false) predictions on each class. Each row of the matrix represents the instances in an actual class, and each column represents the instances in a predicted class, or vice versa. The best accuracy from confusion matrix is 1.0, whereas the worst is 0.0. Classification accuracy alone can be misleading if you have an unequal number of observations in each class, or if you have more than two classes in your dataset. Calculating a confusion matrix can give you a better idea of what your classification model is getting right and what types of errors it is making.

The confusion matrix has 100% accuracy for both classes in test conditions, as seen in the figure below. A classification problem’s prediction outcomes are compiled in a confusion matrix. Count values are used to describe the number of accurate and inaccurate predictions for each class. When it makes predictions, this leads to confusion and is the confusion matrix’s key. The confusion matrix is plotted for 50 images only, where 10 images are extracted for the features vector using the machine learning model based on the algorithm. It is observed that the confusion matrix has 100% accuracy in classifying those 10 images. We have done so in order to save the run time. This explanation has now been included in the manuscript.

### 3.2. Analysis of Survival and Classification of Variables

The results of the analyses for each group can be visualised in survival graphs (Figure 3) where the probability of occurrence of the death event is evaluated over time, and it is possible to visualise in which occurrence the individuals were classified. In the graphs, only individuals not included in the algorithm training process are represented—those whose survival the machine will predict. The following figures show the groups that presented an error rate below 20%. Briefly, the groups that showed good prediction were skin, breast, prostate, cervix, colon and rectum, lymphatic system, and head, face, and neck cancers. Considering the skin cancer group, there is a high probability of survival for all individuals (Figure 3), with few predicted deaths (red lines); 31 being the total predicted deaths in a group of 2038 patients, and the average total probability of survival being above 95%. The group included melanoma and non-melanoma skin cancers.

A statistical tool set called “survival analysis” is used to estimate how long until an event occurs. As the name suggests, this “event” might result in the patient’s survival or the death of humans with a particular disease process. The survival function can be calculated from data that have been censored, truncated, or have missing values using the Kaplan-Meier curve. It displays the likelihood that a subject will live until the end. Plotting the survival function against time results in the creation of the curve. The researcher is frequently curious about how different therapies or predictor variables affect survival. Any point on the survival curve thus represents the likelihood that a patient receiving a specific treatment will not have felt better by that point. The blue line predicts survival, whereas the red lines in the following Figure 3, Figure 4, Figure 5, Figure 6 and Figure 7 indicate predictions of death.

The variables classified as most relevant for predicting the result mentioned above (Figure 3) have assigned importance values. The values numerically indicate the change that the removal of the variable would cause in the error rate, the most important being the one that would add a greater value to this rate [28]. Variables with values below zero indicate that their omission improves prediction performance. With the classification given by the algorithm, the most relevant clinical factors for the prognosis of the disease are identified, considering the regional pattern of the patients.

Age is pointed out as the main indicator within the attributes of patients, which is expected for the type of cancer in question since it is more frequent in older people [26,27]. The variables related to the treatment are also positioned at the top of the graph, indicating the influence of the first treatment used and its progression, demonstrating that there is a positive correlation between these indicators and patient survival in the evaluated group. Then, variables related to the pathological classification of the tumor and staging of the disease are important because they are already consolidated clinical indicators [29]. The survival rate does decrease for later-stage skin cancers, when the cancer has metastasized, or spread, to the lymph nodes or other areas of the body.

Figure 4 shows the same variable classification chart for the breast cancer group. In the data set in question, treatment progression is pointed out as the greatest indicator for predicting survival. It can be considered that the group pattern is mainly determined by the treatment outcome. Advances in cancer treatments in recent years have shown an increase in survival in different age groups [30]. In Iraq, studies evaluating the efficiency of treatments through survival analyses show a pattern of increase in this rate in the country’s most developed regions, where the importance of effective diagnosis and treatment is highlighted [31].

Among the other factors, staging and pTNM are variables considered important for the prognosis of breast cancer in survival analysis in the Southeast region of Iraq. Considered over a period of ten years, using the Cox proportional hazards model, these were also classified as variables related to tumor size.

Regarding the probability of survival for breast cancer (Figure 5), 66 deaths were predicted, with a survival rate above 40% for all patients, with the total average being above 90%. Breast cancer survival rates compare the number of women with breast cancer to the number of women in the overall population in order to estimate the amount of time women with breast cancer are likely to live after they’re diagnosed.

Regarding prognostic factors for prostate cancer, the variables classified as most important were treatment progression, TNM, and pTNM. Diagnostic (TNM and pTNM) and treatment (progression or efficiency) strategies are related to survival when it is desired to evaluate the efficiency of health services applied in a population, leading to an optimisation in the decision-making process for each type of cancer disease. Considering the above predictions, it is possible to verify that tumor staging is a variable of moderate importance. The TNM classification system of malignant tumors is widely applied to perform this classification. For the prostate cancer group (Figure 6), the probability of survival was above 45% for all patients, with the total average being above 90% and with 54 deaths observed according to the forecast. It is important to remember that statistics on the survival rates for people with prostate cancer are an estimate. The estimate comes from annual data based on the number of people with this cancer.

The classification of variables for the rest of the groups with an error rate below 20% followed a pattern where treatment progression was presented as the most important variable, with variation only in the value of time in each group. Regarding the survival rate, the cervical cancer group (Figure 7) had a probability above 50% for all patients, with a total average above 80% and with 29 deaths predicted. Doctors estimate cervical cancer prognosis by using statistics collected over many years from people with cervical cancer. For example, the 5-year relative survival rate for cervical cancer diagnosed at an early stage is 92%. As for head (Figure 7) and face and neck cancer, the probability for all patients was above 25%, with the average above 75% and with 131 deaths predicted. For cancer in the lymph nodes (Figure 7), the probability for all patients was above 45%, and the total mean was above 75% with 44 deaths predicted. In the colon and rectal cancer group (Figure 7), the probability for all patients was above 25%, and the total mean was above 80% with 55 deaths predicted.

Frequent skin cancers may have a survival rate of above 95%. This is because the first biopsy technique is often enough to eliminate the tumor in instances of basal cell and squamous cell carcinoma. However, certain situations do need further medical attention. When skin cancer has metastasized, or spread, to the lymph nodes or other parts of the body, the survival rate does decline.

Breast cancer survival rates calculate the probability that a woman with breast cancer will survive after being diagnosed by comparing the number of cases to the total number of women in the community. For instance, if a stage of breast cancer has a 5-year survival rate of 90%, it indicates that women diagnosed with that stage have a 90% greater chance of surviving for 5 years than women who do not have the disease.

It’s vital to keep in mind that figures on prostate cancer patient survival rates are estimates. The estimate is based on yearly statistics on the prevalence of this malignancy. Additionally, every five years, specialists assess the survival rates. This implies that the estimate could not take into account changes in the previous five years to the diagnosis or treatment of prostate cancer.

Using data gathered from cervical cancer patients over a long period, doctors evaluate the prognosis of the disease. For instance, cervical cancer that is detected early has a 92% relative 5-year survival rate. This indicates that the likelihood of survival five years following diagnosis is 92% higher for those with early-stage cervical cancer than for those without the disease.

According to the study’s findings, the lowest incidence of skin cancer was 7.11% and the highest rate for liver cancer was 65% in Table 1. The numbers demonstrate that the groups who showed an error rate of fewer than 20%. The cohort includes both melanoma and non-melanoma skin malignancies, with just 31 anticipated deaths overall in a group of 2038 patients with an average overall survival probability of above 95%.

## 4. Conclusions

The clinical data from the Hospital Cancer Registries made it possible to perform survival analyses considering all the variables of the patient’s medical records, with relatively high precision for four of the eleven groups evaluated. With the classification of the most important variables for predicting events, the characteristic pattern of the database of patients treated by the Baghdad Teaching Hospital during the period from 2018 to 2021 was observed. The pattern corresponded to a response mainly related to the treatment progression of patients in all groups, except the skin cancer group, where the most important variable corresponded to age. The various ML approaches are related to SVM, 1NN, MLP, and RF. In this work we have used a deep learning model for training and RF for prediction of results. Machine learning techniques allow a global assessment of the available data, showing results that can be interpreted as important information for epidemiological studies, even in cases where the sample is small and there is a lack of information on several variables. The results of the study showed the lowest rate was 7.11% for the skin cancer group, and the highest rate was 65% for the cancer group of the liver (the groups that presented an error rate below 20%).

## Figures and Tables

**Figure 1 healthcare-11-00080-f001:**
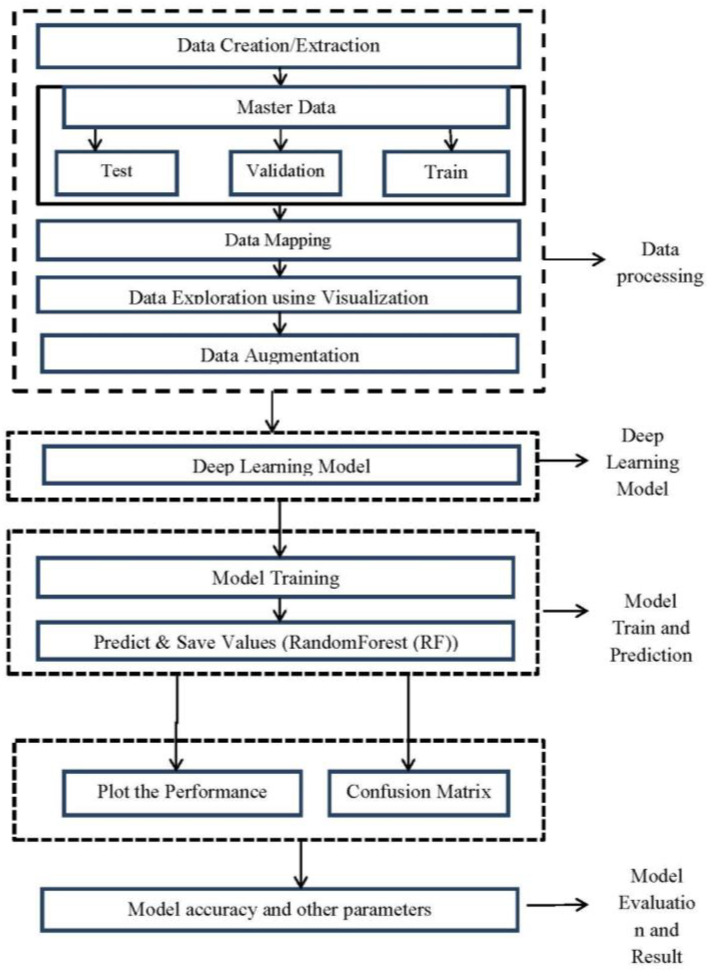
The flow chart for the proposed methodology.

**Figure 2 healthcare-11-00080-f002:**
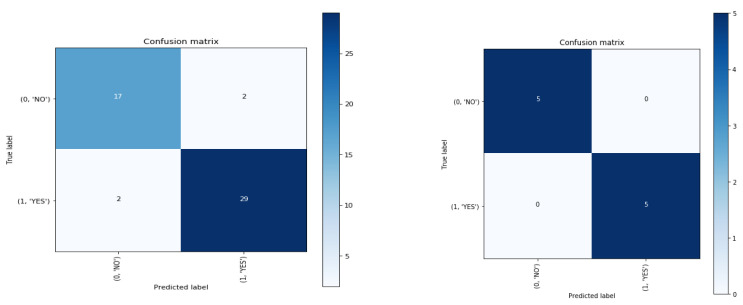
Four different regions (confusion matrix) to provide the extraction of ML model’s feature vector.

**Figure 3 healthcare-11-00080-f003:**
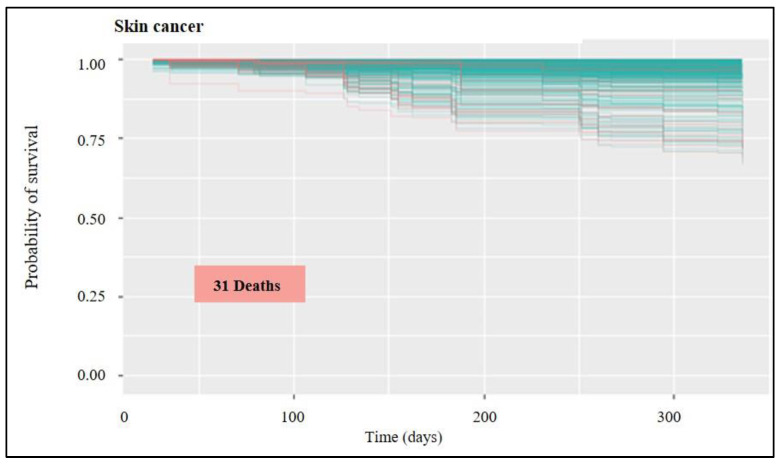
Skin cancer survival graph: probability of survival over time for the untrained group of patients (out-of-bag group). In the red is prediction of death occurrences and in blue is survival.

**Figure 4 healthcare-11-00080-f004:**
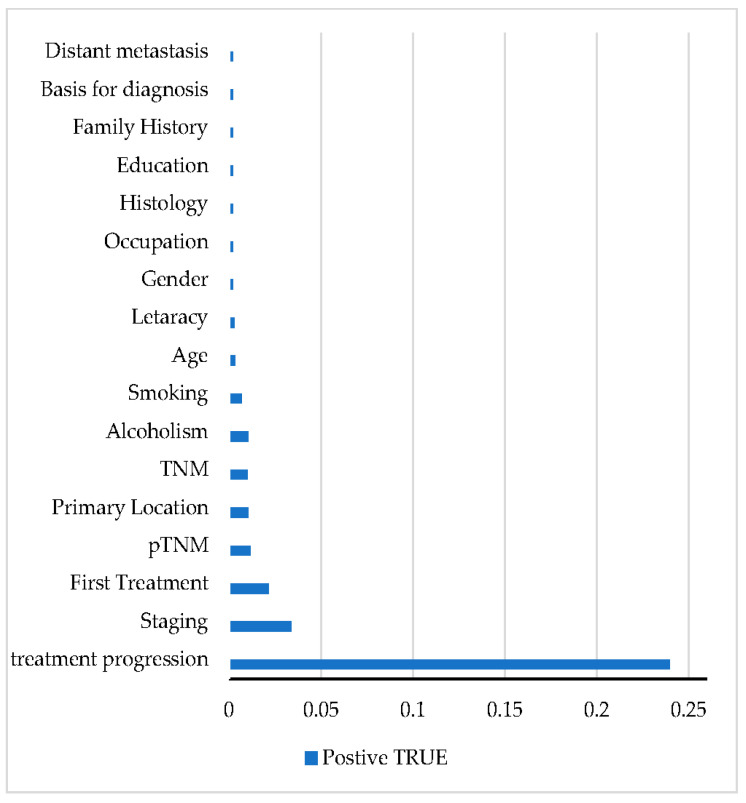
Classification of variables for breast cancer; classification of variables according to the importance for prediction (vimp). Blue bars indicate positive influence; red bars indicate negative influence.

**Figure 5 healthcare-11-00080-f005:**
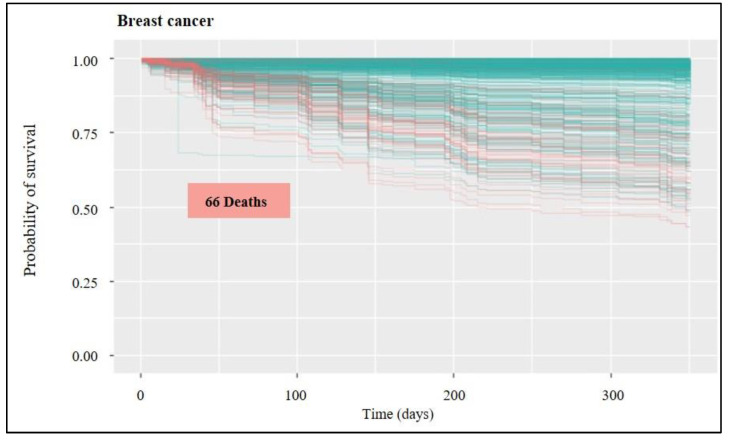
Survival for breast cancer: probability of survival over time for the untrained group of patients (out-of-bag group). In red lines is the forecast of death occurrences; in blue is survival.

**Figure 6 healthcare-11-00080-f006:**
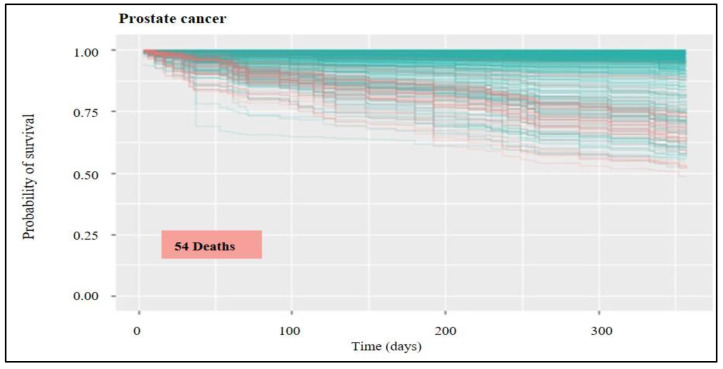
Survival for prostate cancer: probability of survival over time for the untrained group of patients (out-of-bag group). In red lines is the prediction of death occurrences; in blue is survival.

**Figure 7 healthcare-11-00080-f007:**
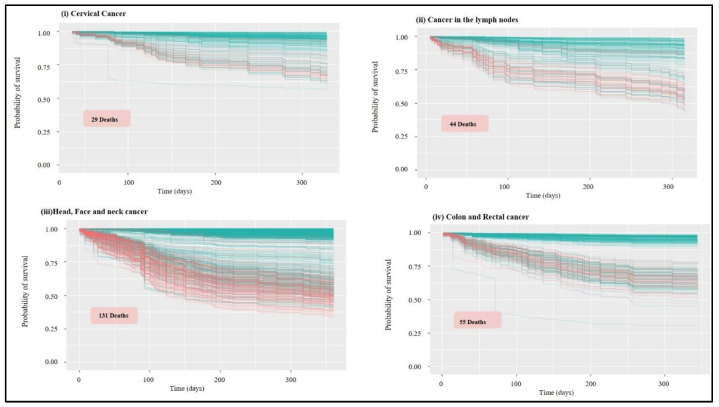
Survival graphs: probability of survival over time for the untrained group of patients (out-of-bag group), representing from top to bottom: (**i**) cervical cancer; (**ii**) cancer of the lymph nodes; (**iii**) cancer of the head, face and neck; (**iv**) colon and rectal cancer. In red is the prediction of death occurrences, and in blue is survival.

**Table 1 healthcare-11-00080-t001:** Prediction error rate: prediction error rate and average Brier Score value for each type of cancer, considering a total of 1000 trees created.

Cancer Location	Dataset Type (Grade Based)	Error Rate	Brier Score	No of Samples (Human)
Head, face, and neck	Different dataset/simple dataset with class column	0.1189	0.1	855
colon and rectum	Different dataset	0.1869	0.1	321
Cervix	Classes are already define in dataset	0.1531	0.09	233
Stomach	Simple dataset	0.2672	0.15	253
Liver	Simple dataset	0.65	0.26	59
lymph nodes	Multi column simple dataset with classes defined	0.1648	0.15	207
mama	Same dataset	0.1014	0.09	981
pancreas	Different Dataset	0.5986	0.25	53
Skin	Complex Dataset	0.0711	0.01	2038
Prostate	Complex Dataset	0.1084	0.05	629
Lung	Simple Dataset	0.3035	0.19	248
	Total = 5877			

## Data Availability

Not applicable.

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
