# Peer review of "Survival Analysis of Oncological Patients Using Machine Learning Method"

_healthcare, 2022, doi:10.3390/healthcare11010080_

Round 1

Reviewer 1 Report

1. There are couple of typos in the manuscript. The whole manuscript needs to be checked once.

2. The presentation of the results could be better, particularly, Figure 2, 3, and 4. It is difficult to differentiate the different lines in the figures, especially the colours. Maybe an alternate technique to present the result could help as the current way is difficult to interpret. 

Reviewer 2 Report

This paper presents a ML-based approach to predict survival of oncological patients based on mutiple information.
Overall comments:
The research work is very useful for the community since it highlights the important factors that contribute in patients survival. However, from the ML perspective, I think this paper lack multiple detail information. Therefore, it is very hard to reproduce the results, and to evaluate how correct the experiments are. Therefore, I believe that major revision is needed to improve the cleanliness and clearness of this paper.

Major issues:
- To correctly present a ML approach and make it reproducible, the authors must provide how the training/testing data is divided, what are the sizes of the training/testing set and the hyperparameters for the ML model (e.g. max depth, max leaf nodes, ... in the case of RF).
- To correctly conduct a ML experiment, evaluation is extremely important. For a binary classification approach, the authors should consider using confusion matrix and k-fold validation.
- From what I understand, the 5877 samples is the training set. However, in table 1 where the authors present the prediction error rate on the same 5877 samples, which makes it hard to follow.
- Figures 1,3,4,5 is badly explained and badly presented. Therefore, it is very hard to understand what these lines represent.

Minor issues (mostly presentation and grammar issues):
- Subsection 2.0 should be named Section 2
- Subsection 2.1 title should be aligned with other subsections
- All tables and figures should have some padding
- Figure 2 instead of Positive FALSE and Positive TRUE, I think it should be Positive and Negative. Furthermore, there is no variable that has negative effect on the prediction score. Therefore, I think the Negative color should be omitted.
- Line 126: Itself (classification of individuals into different levels of risk) - This is not a sentence.

Reviewer 3 Report

Please revise the paper in the following parts:

1) What are the key contributions of the paper? Specify at the end of introduction.

2) Describe organization of the paper too at the end of Introduction.

3) Commentary about the "Related Works" is missing. Incorporate a new section in this regards to justify that how your work is different from the existing works.

4) Incorporate some more results. How these results are verified? Please discuss.

5) Support your conclusion with key results and research insights.

Author Response

Thank you for your comments on our manuscript. We have studied the comments carefully and made corrections which we hope meet with approval. All revised portions are marked in green in the revised manuscript and the responses to the reviewers.

Reviewer 4 Report

I have a few suggestions for the author as follows:

1. The systematical academic writing is an issue in this paper.

2. The paper is not highly analyzing oncological patients as the title.

3. It is not clear what machine learning methods are used, is that a comparison? please give a conclusion of the experimental' result and any research direction.

Author Response

Thank you for your comments on our manuscript. We have studied the comments carefully and made corrections which we hope meet with approval. All revised portions are marked in yellow in the revised manuscript and the responses to the reviewers.

Round 2

Reviewer 2 Report

Major issues: The authors have added information on page 6,7,8. However, some issues still have not been addressed correctly. - Training/testing method is still a huge problem. Even though comments were added on page 6 about how the training/test/validation set is divided, I think it is not the correct way to conduct a ML experiment (the authors proposed multiple ratio and choose the one with best outcome). The correct way is to either use k-fold validation or using a train/test split that is suitable for this unbalanced problem (for example, there are alot of Skin cancer samples and not much Liver cancer samples). The unbalanced dataset is the reason behind the high error rate and Brier score for some cancer types and must be dealt with. - Page 7 line 273-288 is badly written and is very hard to understand. The authors just listed the numbers from Table 1 with no particular reason or comments. - Explaination on confusion matrix (page 7 lines 306-311) is also badly written and provides useless information.

Reviewer 4 Report

Dear author,

The reviewer is trying to find the contribution for this paper. But there are few contributions from there. Is this paper contribution "1. data collection which is useful for other researchers on future" or "2. methodological approach on which the proposed method has strengthen comparing with state-of-the-art methods". Please answer this question with the prove on the manuscript.

If the answer is second option, the reviewer cannot see the contribution on the paper due to the methodology is basic method and the algorithm is available library on the R studio (line 244).
